

# Vertical partitioning of CO$_2$ production in a Dystric Cambisol

Patrick Wordell-Dietrich[1,2], Axel Don[2], Anja Wotte[3,4], Janet Rethemeyer[4], Jörg Bachmann[5],
Mirjam Helfrich[2], Kristina Kirfel[6], and Christoph Leuschner[6]

[1]Institute of Soil Science and Site Ecology, Technische Universität Dresden, Pienner Straße 19, 01737 Tharandt, Germany
[2]Thünen Institute of Climate-Smart Agriculture, Bundesallee 50, 38116 Braunschweig, Germany
[3]Institute of Geology, Technische Univesität Bergakademie Freiberg, Bernhard-von-Cotta Straße 2, 09599 Freiberg, Germany
[4]Institute of Geology and Mineralogy, University of Cologne, Zülpicher Straße 49b, 50674 Köln, Germany
[5]Institute of Soil Science, Leibniz University Hannover, Herrenhäuser Straße 2, 30451 Hannover, Germany
[6]Plant Ecology, Albrecht Haller Institute for Plant Science, University of Göttingen, Untere Karspüle 2, 37073 Göttingen, Germany

**Correspondence:** Patrick Wordell-Dietrich (patrick.wordell-dietrich@tu-dresden.de)

**Abstract.** Large amounts of total organic carbon are temporarily stored in soils, which makes soil respiration one of the major sources of terrestrial CO$_2$ fluxes within the global carbon cycle. More than half of global soil organic carbon (SOC) is stored in subsoils (below 30 cm), which represent a significant C pool. Although several studies and models have investigated soil respiration, little is known about the quantitative contribution of subsoils to total soil respiration or about the sources of CO$_2$
production in subsoils. In a two-year field study in a European beech forest in northern Germany, vertical CO$_2$ concentration profiles were continuously measured at three locations and CO$_2$ production quantified in the topsoil and the subsoil. To determine the contribution of fresh litter-derived C to CO$_2$ production in the three soil profiles, an isotopic labelling experiment using $^{13}$C-enriched leaf litter was performed. Additionally, radiocarbon measurements of CO$_2$ in the soil atmosphere were used to obtain information about the age of the C source in CO$_2$ production. At the study site, it was found that 90 % of total soil
respiration was produced in the first 30 cm of the soil profile where 53 % of the SOC stock is stored. Freshly labelled litter inputs in the form of dissolved organic matter were only a minor source for CO$_2$ production below a depth of 10 cm. In the first two months after litter application, fresh litter-derived C contributed on average 1 % at 10 cm depth and 0.1 % at 150 cm depth to CO$_2$ in the soil profile. Thereafter, its contribution was less than 0.3 % and 0.05 % at 10 cm and 150 cm depths respectively. Furthermore CO$_2$ in the soil profile had the same modern radiocarbon signature at all depths, indicating that CO$_2$
in the subsoil originated from young C sources, despite a radiocarbon age bulk SOC in the subsoil. This suggests that fresh C inputs in subsoils in the form of roots and root exudates are rapidly respired and that other subsoil SOC seems to be relatively stable. The field labelling experiment also revealed a downward diffusion of $^{13}$CO$_2$ in the soil profile against the total CO$_2$ gradient. This isotopic dependency should be taken into account when using labelled $^{13}$CO$_2$ and $^{14}$C isotope data as an age proxy for CO$_2$ sources in the soil.



# 1   Introduction

Soils are the world's largest terrestrial organic carbon (C) pool, with an estimated global C stock of about 2400 Gt in first two metres of the world's soils (Batjes, 2014). The $CO_2$ efflux from soils, known as soil respiration, is the second largest flux component in the global C cycle (Bond-Lamberty and Thomson, 2010; Raich and Potter, 1995) and can be divided

into autotrophic respiration due to roots black and mycorrhizae and heterotrophic respiration due to mineralization of soil organic carbon (SOC) by decomposers. Global warming is expected to increase soil respiration by boosting the microbial decomposition of SOC (Bond-Lamberty et al., 2018; Hashimoto et al., 2015) and by greater root respiration (Schindlbacher et al., 2009; Suseela and Dukes, 2013). Although most of the $CO_2$ is produced in topsoils (< 30 cm), a significant amount of $CO_2$ is produced in the subsoil (> 30 cm) (Davidson and Trumbore, 1995; Drewitt et al., 2005; Fierer et al., 2005; Jassal et al.,

2005). Despite the fact that more than 50 % of global SOC stocks are stored in subsoils (Batjes, 2014; Jobbágy and Jackson, 2000), little is known about the amount and sources of $CO_2$ production in subsoils. Moreover, the mechanisms controlling $CO_2$ production in subsoils are still not fully understood. High apparent radiocarbon ($^{14}$C) ages of SOC in subsoils (Rethemeyer et al., 2005; Torn et al., 1997) lead to an assumption of a high stability of C and a low turnover in subsoils. However, laboratory incubations of subsoil samples show similar mineralisation rates of SOC in both subsoils and topsoils (Agnelli et al., 2004;

Salomé et al., 2010; Wordell-Dietrich et al., 2017), suggesting that subsoils also contain a labile fraction that should be taken into account as a source for soil respiration.

A range of studies have been conducted on $CO_2$ production in soils, but most of them have focused on spatial variations in temperature, water content and substrate supply (Borken et al., 2002; Davidson et al., 1998; Fang and Moncrieff, 2001), but ignoring the vertical partitioning of $CO_2$ production in the whole soil profile which is essential for understanding soil

C dynamics. One reason for this might be the measurement methods used to quantify sources and fluxes in the soil profile. Total $CO_2$ production can easily be measured at the soil surface with an open-bottom chamber, whereas vertical monitoring of $CO_2$ production needs determination of $CO_2$ concentrations at several soil depths in order to estimate $CO_2$ production, i.e. using the gradient method first described by de Jong, E., Schappert (1972). Basically, the $CO_2$ flux between two depths can be calculated using the effective gas diffusion coefficient and the $CO_2$ gradient between the two depths. Recently, the

development of low-cost sensors for temperature, soil moisture and $CO_2$ concentration has allowed greater use of the gradient method (Jassal et al., 2005; Maier and Schack-Kirchner, 2014; Pingintha et al., 2010; Tang et al., 2005). This method can help quantify $CO_2$ production in the entire soil profile, which is essential for an improved quantitative understanding of whole soil C dynamics including the important contribution made by subsoil. To date there have only been a few studies that have continuously determined $CO_2$ production in the whole soil profile *in situ* over a longer timescale (Goffin et al., 2014; Moyes

and Bowling, 2012).

In the present study, the vertical distribution of $CO_2$ concentration was measured and $CO_2$ production rates calculated over a two-year period in a Dystric Cambisol in a temperate beech forest. The objectives of this study were 1) to quantify the contribution of $CO_2$ production in subsoils to total soil $CO_2$ production, and 2) to identify sources of $CO_2$ production along the





soil profile using sources partitioning *via* isotopic data ($^{13}$C and $^{14}$C). It was hypothesised that the majority of $CO_2$ in subsoils originates from young C sources and not from mineralisation of old SOC.

## 2 Methods

### 2.1 Site description and subsoil observatories

The study site is located in a beech forest (Grinderwald) 35 km northwest of Hannover, Germany (52°34´22´´N, 9°18´49´´E). The vegetation is dominated by common beech trees (*Fagus sylvatica*) that were planted in 1916 and the soil is characterised as a Dystric Cambisol (IUSS Working Group WRB, 2014) developed on Pleistocene fluvial and aeolian sandy deposits from the Saale glaciation. The site is located around 100 m above sea level, with a mean annual temperature and precipitation of 9.7 °C and 762 mm (1981–2010) respectively. The soil texture of the site is mainly composed of the sand fraction with contents
varying from 60 % (< 30 cm) to 90 % (> 120 cm), with SOC contents of 11.5 g kg$^{-1}$ down to (10 cm) 0.4 g kg$^{-1}$ (185 cm) (Heinze et al., 2018; Leinemann et al., 2016).

In July 2013, three subsoil observatories were installed using a stainless steel lysimeter vessel (1.6 m diameter and 2 m height) driven 2 m deep into the soil. Once the vessel had been inserted, the soil inside the containment was excavated by hand and undisturbed soil cores (5.7 cm inner diameter, 4.0 cm height) taken with five replicates at depths of 10, 30, 50,
90 and 150 cm from each subsoil observatory for soil diffusivity measurements. In addition, undisturbed soil samples in the observatories were taken to estimate fine root density. Thus six samples were taken from the forest floor and six samples from each of the upper mineral soil layers (0–10 cm, 10–20 cm, 20–40 cm) using a soil corer (3.5 cm diameter), and three samples were taken from each depth increment of the lower profile (40–200 cm depth) at 20 cm depth intervals using a steel cylinder (12.3 cm diameter and 20 cm height). In the laboratory, the samples were gently washed over sieves of 0.25-mm mesh size to
separate the roots from adhering soil particles. Under the stereo microscope, the rootlets were separated into live (biomass) and dead (necromass) roots, and subsequently into fine (< 2 mm in diameter) and coarse roots (> 2 mm in diameter). All live and dead root samples were dried at 70 °C for 48 h and weighed.

After the lysimeter vessel was removed, a polyethylene shaft (1.5 m in diameter and 2.1 m height) was placed in the soil, referred to here as the subsoil observatory. The gap (≈ 5 cm) between the subsoil observatory and the surrounding undisturbed
soil was refilled. The observatories where installed close to one other, with a maximum distance of 30 m between them.

To monitor the temperature and volumetric water content, combined temperature and moisture sensors (UMP-1,Umwelt-Geräte-Technik GmbH, Germany) were installed at depths of 10, 30, 50, 90 and 150 cm with a horizontal distance of 100 cm from the wall of the subsoil observatories. Measurements were taken every 15 minutes and stored on a data logger. The $CO_2$ concentration in the soil air was monitored by solid-state infrared gas sensors (GMP221, Vaisala Oyi, Finland) with a
measuring range of 0–10 % $CO_2$. To protect the PTFE membrane of the $CO_2$ sensor from damage while being placed in the soil, the sensor was coated with an additional PTFE foil (616.13 P, FIBERFLON, Turkey), to allow gaseous diffusion and prevent water infiltration. The $CO_2$ concentration was measured every three hours to reduce power consumption. The $CO_2$ sensors were turned on 15 minutes before the measurement itself due to their warm-up time. In addition, PTFE suction cups





for soil air sampling with stainless steel tubing (ecoTech Umwelt-Meßsysteme GmbH, Germany) were installed adjacent to the $CO_2$ sensors. The gas samplers and $CO_2$ sensors were installed at the same depths as the temperature and moisture sensors. The horizontal distance of the gas samplers and $CO_2$ sensors from the subsoil observatory wall increased from 40 cm to 100 cm with increasing soil depth.

## 2.2 Gas sampling and measurements

### 2.2.1 Soil respiration

The surface $CO_2$ efflux was measured using the closed-chamber method. Thirty PVC collars with a diameter of 10.4 cm and a height of 10 cm were installed 5 cm deep in the soil around the three subsoil observatories. The organic layer of 15 collars was removed in order to be able to distinguish between mineral soil respiration and total soil respiration. Soil respiration was measured with the EGM-3 SRC-1 soil respiration chamber (PP-Systems, USA) and the LI-6400-09 soil chamber (LI-COR Inc., USA). The measurement system was changed due to technical problems with the EGM-3 system, however a comparison between the two systems revealed only minor differences. Each collar was measured three times per sampling day from March 2014 to March 2016, with sampling ranging from once a month to once a week. Annual soil respiration was derived from linear interpolation of measured $CO_2$ fluxes from the collars. Furthermore, soil respiration was modelled by fitting an Arrhenius-type model (Eq.1), introduced by Lloyd and Taylor (1994) and using soil temperature data from 10 cm depth, and the measured $CO_2$ fluxes:

$$F_0 = a \times e^{\left( \frac{E_0}{T + 273.2 - T_0} \times \frac{T - 10}{283.2 - T_0} \right)} \tag{1}$$

where $F_0$ is soil respiration [$\mu$mol m$^{-2}$ s$^{-1}$], $a$, $E_0$ and $T_0$ are fitted model parameters, and $T$ is the soil temperature at 10 cm depth [°C].

### 2.2.2 $^{13}CO_2$ sampling and measurement

In addition to continuous $CO_2$ concentration monitoring, two gas samples per depth and subsoil observatory were taken from the suction cups with a syringe and filled into 12-mL evacuated gas vials (Labco Exetainer, Labco Limited, UK). The sampling started in May 2014 with an interval of between once a month and once a week. The $CO_2$ concentration in the soil gas samples was analysed by gas chromatography (Agilent 7890A, Agilent Technologies, USA). The $\delta^{13}C$ values of the $CO_2$ samples were measured by an isotope ratio mass spectrometer (Delta Plus with GP interface and GC-Box, Thermo Fisher Scientific, Germany) connected to a PAL autosampler (CTC Analytics, Switzerland). The $^{13}C$ results are expressed in parts per thousand (‰) relative to the international standard Vienna Pee Dee Belemnite (VPDB).

### 2.2.3 $^{14}CO_2$ sampling and measurement

Soil gas samples for radiocarbon analysis were taken in October and December 2014 in subsoil observatories 1 and 3. The $CO_2$ was sampled using a self-made molecular sieve cartridge as described in Wotte et al. (2017). Briefly, each stainless steel



cartridge was filled with 500 mg zeolite type 13X (40/60 mesh, Charge 5634, IVA Analysetechnik GmbH & Co KG, Germany), which is used as an adsorbent for $CO_2$. The molecular sieve cartridges were connected to the installed gas samplers. The soil atmosphere of the corresponding depth was then pumped with an airflow of 7 mL min[1] over a desiccant (Drierite, W. A. Hammond Drierite Company, USA) to the molecular sieve cartridge for 40 minutes to trap the $CO_2$ on the molecular sieve.

Surface samples were taken from a respiration chamber (Gaudinsik et al., 2000). The atmospheric $CO_2$ inside the chamber was removed prior to sampling by circulating an airflow of $\approx 1.5$ L min[-1] from the chamber through a column filled with soda lime until the equivalent of 2-3 chamber volumes had been passed over the soda lime. Thereafter, the airflow was run over a desiccant and the molecular sieve cartridge for 10 minutes to collect the $CO_2$ sample.

In the laboratory, the adsorbed $CO_2$ was released from the molecular sieve cartridge by heating the molecular sieve under

vacuum (Wotte et al., 2017). The released $CO_2$ was purified cryogenically and sealed in a glass tube. The radiocarbon ([14]C) analysis was directly performed on the $CO_2$ with the gas ion source of the mini carbon dating system (MICADAS, Ionplus, Switzerland) at ETH Zurich (Ruff et al., 2010). The [14]C concentrations are reported as fraction modern carbon (F[14]C), whereby F[14]C values less than one denote that the majority of the C was fixed before the nuclear bomb tests in the 1960s, while values greater than one indicate C fixation after the bomb tests.

## 2.3   Labelling experiment

To trace the fate of fresh litter inputs in the soil and their contribution to the $CO_2$ released from different soil horizons, a [13]C labelling experiment was performed. In January 2015, the leaf litter layer around the subsoil observatories was removed and replaced with a homogeneous mixture of 237 g [13]C-labelled and 1575 g non-labelled young beech litter, which is equal to a litter input of 250 g m[-2]. The labelled litter was distributed on a semi-circular area around the subsoil observatories.

The labelled litter originated from young beech trees grown in a greenhouse in a $^{13}CO_2$-enriched atmosphere. The mixture of labelled and non-labelled litter had an average $\delta^{13}C$ value of 1241 ‰ for subsoil observatory 1 (OB1) and a $\delta^{13}C$ value of 1880 ‰ for subsoil observatories 2 (OB2) and 3 (OB3).

## 2.4   Diffusivity measurements

Gas transport along the soil profile is determined by the diffusivity of the soil. The diffusivity of the soil was determined at

depths of 10, 30, 50, 90 and 150 cm, with five undisturbed core sample replicates per depth and per observatory. To account for different water contents, the undisturbed soil cores (5.7 cm diameter, 4.0 cm height) were adjusted in the laboratory at different matrix potentials (-30 hPa, -60 hPa, -300 hPa) to cover a wide range of soil moisture. After moisture adjustment, the soil cores were attached to a diffusion chamber as described in Böttcher et al. (2011). The diffusion chamber was flushed with $N_2$ to initially establish a gas gradient between the chamber and the top of the sample as an atmospheric boundary condition.

The increase in oxygen inside the ventilated chamber was measured over time with an oxygen dipping probe (DP-PSt3-L2.5-St10-YOP, PreSens-Precision Sensing GmbH, Germany). Diffusivity and tortuosity factors ($\tau$) were calculated with an inverse diffusion model (Schwen and Böttcher, 2013).



## 2.5 Data analysis

### 2.5.1 Gradient method

This method is based on the assumption that molecular diffusion is the main gas transport in the soil atmosphere. Therefore gas fluxes, e.g. $CO_2$ fluxes in a soil profile, can be calculated from the $CO_2$ concentration gradient and the effective gas diffusion

coefficient in the specific soil layer of interest.

In order to account for temperature and pressure dependencies of the $CO_2$ sensors, the $CO_2$ concentrations were corrected with a compensation algorithm for the GMP221 (S1) provided by the manufacturer (pers. comm. Niklas Piiroinen, Vaisala Oyi, Finland). For the flux calculation, $CO_2$ volume concentrations were converted to $CO_2$ mole concentrations (2):

$$C = \frac{C_v \times p}{R \times T} \tag{2}$$

where $C$ is the $CO_2$ mole concentration [μmol m$^{-3}$], $C_v$ is the $CO_2$ volume fraction [μmol mol$^{-1}$], $p$ is the atmospheric pressure in [Pa], $R$ is the universal gas constant [8.3144 J K$^{-1}$ mol$^{-1}$] and $T$ is the soil temperature in [K] measured by temperature sensors at the corresponding soil depths. The $CO_2$ flux of a soil layer was calculated using Fick's first law (Eq. 3)

$$F = -D_s \times \frac{dC}{dz} \tag{3}$$

where $F$ is the diffusive $CO_2$ flux [μmol m$^{-2}$ s$^{-1}$], $D_s$ is the effective diffusivity in the soil atmosphere [m$^2$ s$^{-1}$] determined as described below, $C$ is the $CO_2$ concentration [μmol m$^{-3}$] and $z$ is the depth [m]. The equation is based on the assumption that 1) molecular diffusion is the dominating transport process in the soil atmosphere and other transport mechanisms – i.e. convective $CO_2$ transport due to air pressure gradients or diffusion in the soil, and convective transport with soil water – are negligible and 2) gas transport is one-dimensional (e.g., de Jong, E., Schappert, 1972; Maier and Schack-Kirchner, 2014). The

effective diffusivity $D_s$ was calculated with Eq. 4:

$$D_s = D_0 \times \tau \tag{4}$$

where $D_0$ is the $CO_2$ diffusivity in free air. The pressure and temperature effect on $D_0$ were taken into account by:

$$D_0 = D_{a0} \times \left(\frac{p_0}{p}\right) \times \left(\frac{T}{T_0}\right)^{1.75} \tag{5}$$

where $D_{a0}$ is a reference value of $D_0$ at standard conditions ($1.47 \times 10^{-5}$ m$^2$ s$^{-1}$ at $T_0$ 293.15 K and $p_0$ $1.013 \times 10^5$ Pa) (Jones,

1994). The dimensionless tortuosity factor $\tau$ at each depth was modelled as a function of the air-filled pore space $\varepsilon$ for each soil depth. The model was derived from a power function fit from laboratory diffusion experiments (see above) on the undisturbed soil cores.

To account for the non-uniform vertical distribution of soil water content in the soil profile, $D_s$ was estimated as the harmonic average between the two measurement depths (Pingintha et al., 2010; Turcu et al., 2005):

$$D_s = \frac{\Delta z_1 + \Delta z_2}{\frac{\Delta z_1}{D_{sz_1}} + \frac{\Delta z_2}{D_{sz_2}}} \tag{6}$$



where $\Delta z_{1,2}$ [m] is the thickness of the corresponding soil layer and $D_{sz_{1,2}}$ is the effective diffusivity of the respective soil layer. Finally, assuming a constant flux between measured $CO_2$ at depth $z_i$ and $z_{i+1}$, the $CO_2$ flux ($F_i$) was calculated by combining Eq. (2 - 6)

$$F_i = \left( \frac{\Delta z_i + \Delta z_{i+1}}{\frac{\Delta z_i}{D_{sz_i}} + \frac{\Delta z_{i+1}}{D_{sz_{i+1}}}} \right) \times \left( \frac{C_{i+1} - C_i}{z_{i+1} - z_i} \right) \tag{7}$$

where $F_i$ is the $CO_2$ flux [µmol m$^{-2}$ s$^{-1}$] at the upper boundary ($z_i$) between depth $z_i$ and $z_{i+1}$[m]. To calculate soil respiration ($F_0$) at the surface with the gradient method, a $CO_2$ concentration of 400 µmol mol$^{-1}$ at the soil surface and a constant $D_s$ for the first 10 cm were assumed.

### 2.5.2   $CO_2$ production

The $CO_2$ production ($P_i$) in a soil layer was calculated as the difference between the flux ($F_i$) leaving the specific soil layer at

the upper boundary ($z_i$) and the input flux ($F_{i+1}$) at the lower boundary ($z_{i+1}$) of the specific soil layer. Therefore, $P_i$ had the unit of a flux [µmol m$^{-2}$ s$^{-1}$].

$$P_i = F_i - F_{i+1} \tag{8}$$

Total soil respiration was calculated as the sum of $CO_2$ production in all soil layers. Equation (8) is based on the assumption of steady-state diffusion. Steady-state conditions for $CO_2$ concentration and volumetric water content were mostly given, except

during a few heavy rain events where steady-state conditions were not met due to changing water contents in the profiles. Most soils exhibit increasing $CO_2$ concentrations with increasing soil depth. Therefore, $CO_2$ production is mostly positive with upward $CO_2$ fluxes. However, if the $CO_2$ concentration in a soil layer is greater than in the layers below, the calculated $CO_2$ production in the layers below can become negative (downward directed). Hence in the present study no $CO_2$ production was assumed when the calculated $CO_2$ production in a soil layer was negative. This approach was based on the assumption that

there are no relevant $CO_2$ sinks in the soil profile. Furthermore, negative $CO_2$ production is considered as $CO_2$ storage, which will be released if the $CO_2$ concentration gradient or diffusion conditions change. In OB1 negative $CO_2$ production values were calculated in the first year at 30-50 cm depth (331 out of 365) and at 50-90 cm depth (359 out of 365). In the second year negative values also occurred in OB1 at 30-50 cm depth (8 out of 308) and at 50-90 cm depth (182 out of 308).

### 2.5.3   Isotopic composition of $CO_2$

To determine the contribution of labelled leaf litter to $CO_2$ in different soil layers, the fluxes of $^{12}CO_2$ and $^{13}CO_2$ had to be calculated separately. Therefore, the amount of $^{13}CO_2$ ($L$) originating from the labelled leaf litter was calculated using the isotopic mixing equation (Eq. 9):

$$L = 1 - \left( \frac{\delta^{13}C_M - \delta^{13}C_L}{\delta^{13}C_B - \delta^{13}C_L} \right) \tag{9}$$

where $\delta^{13}C_M$ is the isotopic signature of the gas sample, $\delta^{13}C_L$ is the isotopic signature of the labelled leaf litter and $\delta^{13}C_B$ is

the average isotopic signature of the gas samples before the labelled leaf litter was applied. The $^{13}CO_2$ volume concentration




for each layer was calculated using Eq. (2) multiplied by $L$. The $^{13}CO_2$ fluxes and production rates were calculated using Eq. (3)-(8). To account for different effective diffusivities of $^{12}CO_2$ and $^{13}CO_2$, the effective diffusivity $D_s$ for $^{13}CO_2$ was adjusted according to Cerling et al. (1991):

$$D_s = {}^{12}D_s = 1.0044 \times {}^{13}D_s \qquad (10)$$

where it is assumed that $D_s$ is equivalent to $^{12}D_s$ due to the fact that about 99 % of total $CO_2$ is $^{12}CO_2$.

## 2.6 Statistical analysis

A Monte Carlo simulation was generated to determine the influence of measurement uncertainties of the sensors, which were used for calculation of $CO_2$ fluxes and $CO_2$ production rates . It was assumed that each measurement error was normally distributed. The standard deviation was equal to measurement accuracy, which was obtained from the corresponding manual.

To obtain a distribution of the power function ($D_s$ model), the Markov chain Monte Carlo algorithm DiffeRential Evolution Adaptive Metropolis (DREAM) (Vrugt et al., 2009) in the R package dream (Guillaume and Andrews, 2012) was used. Dream was run in the standard configuration and as soon as the convergence criteria of Gelman and Rubin (1992)were less than 1.01, another 20000 simulations were run to get a distribution of the $D_s$ model parameters (n=1000). The distributions of $CO_2$, volumetric water content and temperature measurements and the distribution of the $Ds$ model were used for 1000 Monte Carlo

simulations. Unless stated otherwise, the error bars in the final results represent the standard deviation of these simulations. All analyses were performed in R (version 3.3.2) for Linux (R Core Team, 2017).

## 3 Results

### 3.1 Temperature, water content and $CO_2$ concentration in the profile

Soil temperature showed a distinct seasonality down to 150 cm, with the maximum and the minimum temperatures delayed

with increasing soil depth (Fig. 1a). The minimum soil temperature was 0.3 °C and 4.0 °C in January 2016 at 10 cm and 150 cm depths respectively. The maximum temperature was measured in July in the uppermost layer (16.6 °C) and in August in the deepest layer (14.4 °C). The annual amplitude of soil temperature decreased from 16.3 °C at 10 cm to 10.4 °C at 150 cm. However, mean annual values showed no significant decline with soil depth and were 8.4 °C and 8.3 °C at 10 cm and 150 cm respectively during the two years of observation. Variations in the mean soil temperatures between the three observatories were

< 1 °C at all depths (Fig. S1).

The volumetric water contents also showed seasonal variations at all depths (Fig. 1b), with depletion during the summer. The minimum of volumetric water content at 10 cm was reached in August (10 %), whereas the minimum at 150 cm was observed two months later in October (6 %). The water reservoir of the soil profile was refilled during the autumn and winter, reaching maximum values at 10 cm (23 %) and 150 cm (22 %) in April (Fig. 1b), which were delayed by 14 days in the deepest layer.

In OB1 and OB3, the mean volumetric water content decreased with increasing soil depth. Only in OB2 did the mean water



content increase at 150 cm (Fig. S2). The water content showed a greater variation between the three observatories than soil temperature (Fig. S2).

The $CO_2$ concentration in the soil pores followed a similar seasonality as soil temperature (Fig. 1c), with a maximum during the summer and a minimum during the winter and early spring. The same behaviour was observed for both investigated years,

while the values were higher during the first summer. The $CO_2$ concentration in the uppermost layer ranged from 1,000 to 35,000 µmol mol$^{-1}$ and thus was in a similar range of results for the deepest layer with 7,500 to 35,000 µmol mol$^{-1}$. However, values were highly variable between the observatories, with OB2 and OB3 showing an increasing $CO_2$ concentration with greater soil depth, whereas OB1 yielded the highest $CO_2$ concentrations at 30 to 50 cm depth.

## 3.2   Soil respiration

The mean annual mineral (without the organic layer) soil respiration determined with chamber measurements for the three observatories was 776 ± 193 g C m$^{-2}$ yr$^{-1}$, with a small variability between the observatories (Table 1). The mineral soil respiration modelled with the Lloyd-Taylor function gave similar results for the same period. In contrast, soil respiration determined with the gradient method showed a high variability between the observatories, but was in the range of the directly measured respiration, except for OB1. This variability can be explained by the higher water content at OB1 and consequently

the lower diffusion coefficient. The average diffusion coefficient at OB1 at 10 cm was less than half that at OB2 and OB3.

The organic layer increased total respiration by 13 % and 25 % respectively for the Lloyd-Taylor model and chamber measurements (Table 1). For all the methods and in all the observatories, soil respiration correlated well with soil temperature and soil moisture. The highest fluxes were measured when soil temperature (10 cm) was highest and water content (10 cm) was low (Fig. 1 and Fig. 2).

## 3.3   Vertical $CO_2$ production

The mean $CO_2$ production rates decreased from 1.4 µmol m$^{-2}$ s$^{-1}$ in the uppermost layer (0–10 cm depth) to 0.03 µmol m$^{-2}$ s$^{-1}$ in the deepest layer (50–90 cm depth) (Fig. 3). The $CO_2$ production followed the same seasonality as soil temperature and $CO_2$ concentration, with the highest productions rates occurring during the summer and the lowest during the winter months in all soil layers. This seasonal variation was greatest in the top two layers of the soil (0–10, 10–30 cm) (Fig. 3a-d).

About 71 ± 10 % of total soil respiration was produced in the first 10 cm of the soil profile where 21 % of the SOC stock (0–1.5 m) was stored. The $CO_2$ production at 10 to 30 cm accounted for 18 ± 11 % of total soil respiration during the year, and 32 % of the SOC was located in this depth increment. The subsoil (> 30 cm) accounted for 10 ± 8 % of total $CO_2$ production, with 47 % of the SOC stock stored in the subsoil.

The mean total $CO_2$ production showed no significant differences between the two years. The variation in cumulative annual

$CO_2$ production was greater between the three observatories (335–1,203 g $CO_2$-C m$^{-2}$ yr$^{-1}$) than between the two studied years (Fig. 4). However, the $CO_2$ production in the different soil layers showed considerable changes with time: it increased by 500 % in the subsoil from 30 to 50 cm in the second year, which increased the contribution of subsoil $CO_2$ production from 3 % to 15 % of total $CO_2$ production. This increase was observed in all three observatories. In contrast, the $CO_2$ production in the




first 10 cm in OB1 and OB3 showed a decline from the first to the second year, which was probably caused by methodological variations and does not represent a real decrease in respiration activity since bioturbation of animals (e.g. voles) might have had a strong influence on diffusivity (Fig. 3a). Voles created macropores, therefore the $CO_2$ gradient approach was not applicable. This was also indicated by a sudden and rapid drop of $CO_2$ production between 0 and 10 cm in OB1 (October 2015) (Fig. 3a).

5     To take the different SOC contents of each soil layer into account, the cumulative $CO_2$ production was normalised to the SOC stock of the respective layer (Fig. 5). The specific $CO_2$ production decreased from 346 g $CO_2$-C $kg^{-1}$ SOC $yr^{-1}$ in the first 10 cm to less than 8 g $CO_2$-C $kg^{-1}$ SOC $yr^{-1}$ at 50 to 90 cm. It should be noted that the proportion of autotrophic respiration in the total $CO_2$ production could not be quantified.

### 3.4   Sources of $CO_2$ production

10 ### 3.4.1   Contribution of fresh litter

The isotopic signature of soil $CO_2$ ($\delta^{13}CO_2$) in the observatories before the start of labelling experiment ranged from -25.4 ‰ to -21.8 ‰, with no significant differences between soil depths (Fig. 6a). The labelling experiment was conducted to assess the fate of fresh litter added on top of the organic layer into different C fractions (e.g. SOC and DOC) including soil $CO_2$. Six days after the application of the $^{13}$C-labelled leaf litter, $CO_2$ was already enriched in litter-derived C down to 90 cm depth in 15 all the observatories. The isotopic signature ranged from 70 ‰ at 10 cm depth to -19 ‰ at 90 cm depth (Fig. 6b). Thus, the maximum contribution of litter-derived C to total $CO_2$ was 5 % at 10 cm depth six days after the litter replacement (Fig. 6c). At 90 cm, the maximum amount of litter-derived $CO_2$ was 0.6 % two weeks after the beginning of the labelling experiment (Fig. 6c). In addition, minor peaks with up to 0.8 % of $CO_2$ derived from the labelled litter were observed at all depths after rain events within the first six months of litter application. However, the average contribution of litter-derived $CO_2$ decreased 20 with time and reached a range of 2.5 % to 0.2 % at 10 cm depth from January 2015 to July 2016.

    Assuming that diffusion is the main transport process of $CO_2$ in the soil atmosphere, the litter-derived $CO_2$ flux between two soil layers can be calculated according to Eq. (3-7) and Eq. (10). As already mentioned, a positive flux indicates mineralisation of litter-derived C in the respective soil layer. A negative flux in turn represents downward diffusion of litter-derived $CO_2$ from the layer above (Fig. 7). On average for the three observatories, 34 out of 41 sampling days had negative $^{13}CO_2$ fluxes, indicating 25 a downward movement of labelled litter-derived $CO_2$. Only OB1 had positive $^{13}CO_2$ fluxes at 10 to 50 cm, representing a transport of labelled litter-derived C down the soil profile as dissolved organic carbon (DOC) and mineralisation of this DOC. The observed $^{13}$C enrichment in $CO_2$ in OB2 and OB3 was due to diffusion of labelled litter-derived $CO_2$ from the organic layer down to deeper layers of the mineral soil.

### 3.4.2   Contribution of old C

30 The radiocarbon content of the bulk SOC decreased strongly with increasing soil depth from close to atmospheric values ($F^{14}$C 0.99) at 10 cm to an apparent age of about 3460 years BP ($F^{14}$C 0.65) at 110 cm depth (Fig. 9, grey triangles). In contrast, the $^{14}$C concentrations of the $CO_2$ in the soil atmosphere were relatively constant throughout the soil profile and for





both samplings, with values in the range of 1.03–1.07 F$^{14}$C and thus derive mainly from the post-bomb period (Fig. 8, black dots). This indicates a young source of $CO_2$ production. Consequently "old" subsoil SOC was not detected as a significant source of $CO_2$ production.

## 4 Discussion

### 4.1 Temperature, water content and $CO_2$ concentration in the profile

In all three subsoil observatories, increasing $CO_2$ concentrations with depth were observed. This has also been reported by other studies (Davidson et al., 2006; Drewitt et al., 2005; Fierer et al., 2005; Hashimoto et al., 2007; Moyes and Bowling, 2012). However, the increase was not continuous down to 150 cm depth. Higher $CO_2$ concentrations were observed between 30 cm and 50 cm depth, indicating a higher $CO_2$ production at this depth increment, which can be linked to the root distribution in the subsoil observatories (9). About 82 % of the fine root biomass and necromass were found to be located between 0 and 50 cm, and 18 % at the 30 to 50 cm depth. Therefore, the contribution of autotrophic respiration to $CO_2$ production and the mineralisation of dead roots were greater at these depths than in the deep subsoil (> 50 cm). The $CO_2$ concentration in the soil pores is also controlled by abiotic factors such as effective diffusivity ($D_s$). The average effective diffusivity ($D_s$) at 10 cm was about 40 % lower than at 30 cm. Consequently $CO_2$ accumulated in the soil pores below 10 cm depth due to the lower diffusion of $CO_2$ between the soil surface and 10 cm depth. The effective diffusivity was mainly controlled by soil water content, which reduced it. For example, the high $CO_2$ concentration in August 2014 (up to 40,000 µmol mol$^{-1}$) compared to August 2015 (up to 20,000 µmol mol$^{-1}$) (1c) can be explained by the higher volumetric water content in 2014 in all profiles. The high water content was related to more precipitation in July 2014 (120 mm) than in July 2015 (47 mm) and to less precipitation in August in both years (49 and 95 mm). Additionally, evapotranspiration was greater in August 2015 than in August 2014 due to a higher mean air temperature (18 °C and 15 °C).

### 4.2 Soil respiration

The annual mean total respiration determined using the gradient method corresponded well with the results of the closed chamber measurements, indicating that the gradient method resulted in realistic flux estimations (Table 1, Fig. 2). This is in line with the results reported by other studies (Baldocchi et al., 2006; Tang et al., 2003; Liang et al., 2004). The differences in soil respiration between the methods can be attributed to the different spatial resolution of the corresponding measurements. The chamber measurements were based on five spatial replicates for each subsoil observatory, covering a total measurement area of 1274 cm$^2$. Therefore chamber measurements accounted for spatial variability in water content and soil $CO_2$ concentrations below the chamber, whereas the gradient method was based on one profile measurement for $CO_2$ and water content at each of the three observatories. Large differences in total respiration rates of up to 200 % were found between the three observatories with the gradient method. Both methods have advantages and disadvantages for determining total soil respiration. The gradient method does not alter the soil atmosphere $CO_2$ gradient and is continuous and less time-consuming than chamber measure-





ments, but it is very vulnerable to the spatial heterogeneity of the soil structure and moisture content around the sensors and to changes in diffusivity, e.g. due to bioturbation by animals such as voles, which may also led to an underestimation of total soil respiration (e.g. OB1 Fig. 3a).

### 4.3 Vertical $CO_2$ production

The vertically partitioned $CO_2$ flux revealed that more than 90 % of total $CO_2$ efflux was produced in the topsoil (< 30 cm). These results correspond well with other studies which have found that more than 70 % of total $CO_2$ efflux in temperate forests is produced in the upper 30 cm of the soil profile (Davidson et al., 2006; Fierer et al., 2005; Hashimoto et al., 2007; Jassal et al., 2005; Moyes and Bowling, 2012). However, only 53 % of the SOC stock is stored in the first 30 cm, indicating that subsoil SOC on the site of the present study may have a slower turnover than topsoil SOC. This is supported by the low [14]C

concentrations in SOC below 30 cm. However, the higher $CO_2$ production in the topsoil can be also related to greater fine root biomass and necromass density (Fig. 9, which may serve as an indicator of autotrophic respiration and heterotrophic respiration in the rhizosphere. Consequently root-derived respiration is greater in the topsoil than in the subsoil.

   It is remarkable that the $CO_2$ production at 30 to 50 cm increased from 23 g C m$^{-2}$ yr$^{-1}$ in the first year to 118 g C m$^{-2}$ yr$^{-1}$ in the second year of the study (Fig. 4). This can be explained in part by more precipitation in the second year (621 mm) than in the

first year (409 mm), inducing less water-limiting conditions for plants and microbial activity. As a result, the mean volumetric water content was higher in the second year (18 % compared to 16 %) at 50 cm depth, which gave better conditions for the mineralisation of SOC by microorganisms (Cook et al., 1985; Moyano et al., 2012). Furthermore, the greater precipitation increased the input of DOC into the subsoil on the site of the present study, which is supported by the study of (Leinemann et al., 2016) who investigated DOC fluxes in subsoil observatories for more than 60 weeks. They found a positive correlation

between DOC fluxes, precipitation and water fluxes at 10, 50 and 150 cm depths. Furthermore, they showed that DOC fluxes declined by 92 % between a depth of 10 cm and 50 cm, which was attributed to mineral adsorption and microbial respiration of DOC (Leinemann et al., 2016).

### 4.4 Sources of $CO_2$ production

#### 4.4.1 Young litter derived $CO_2$

In this study, a unique labelling approach was used to estimate the contribution of aboveground litter to $CO_2$ production along a soil profile by applying stable isotope-enriched leaf litter to the soil surface. These results showed that litter-derived C did not significantly contribute to annual $CO_2$ production below 10 cm depth. Leaf litter is decomposed and washed into the mineral soil as DOC. Within one year, only 0.2 % of total $CO_2$ production between 10 and 50 cm originated from the labelled leaf litter. Below 50 cm there was no contribution of litter-derived C to $CO_2$ production. Therefore, mineralisation of DOC originating

from the organic layer was a minor source of $CO_2$ production in the soil profile below 10 cm. The average DOC flux in the subsoil observatories in the first year was estimated to be 20 g C m$^{-2}$ yr$^{-1}$ at 10 cm depth and 2 g C m$^{-2}$ yr$^{-1}$ at 50 cm depth, indicating a DOC input of 18 g C m$^{-2}$ yr$^{-1}$ into the 10 and 50 cm depth increments (Leinemann et al., 2016). An assumed





complete mineralisation of this DOC would account for 11 % of $CO_2$ production at this depth increment. Overall, most of the $CO_2$ production between a depth of 10 cm and 50 cm must be derived from autotrophic respiration and heterotrophic respiration in the rhizosphere.

### 4.4.2 Old SOC derived $CO_2$

The very similar radiocarbon contents of soil $CO_2$ produced at different depths, which were 1.06 $F^{14}C$ on average, revealed that ancient SOC components were not a major source of $CO_2$ production. The results indicate that the $CO_2$ originated mainly from young (several decades old) C sources, presumably mainly from root respiration, its exudates and DOC. Other studies have found similar results on a grassland site in California down to 230 cm depth (Fierer et al., 2005) and in temperate forests down to 100 cm (Gaudinsik et al., 2000; Hicks Pries et al., 2017). In addition, Hicks Pries et al. (2017) incubated root-free soil

from three depths (15, 50 and 90 cm) and compared the radiocarbon signature of the respired $CO_2$ with their results from the field. They found that $CO_2$ from the short-term incubations had the same modern signature as the field measurements, despite the high $^{14}C$ age of the bulk SOC at 90 cm depth (~1000 yr BP) (Hicks Pries et al., 2017). This supports the findings of the present experiment. Therefore, microbial respiration in temperate subsoils is mainly fed by relatively young C sources fixed less than 60 years ago.

### 4.4.3 Diffusion effects

A highly $^{13}C$-enriched $CO_2$ source was introduced to the top of a soil profile. Shortly afterwards the application an enrichment of $^{13}C$ was measured in $CO_2$ along the whole soil profile (Fig. 6b). However, this enrichment could not be linked to the transport and mineralisation of litter-derived C along the soil profile (e.g. DOC in seepage water). In contrast, diffusion of $^{13}CO_2$ was observed to have originated from the mineralisation in the litter layer down the soil profile. According to Fick's first law,

$^{13}CO_2$ diffuses into the soil profile following the $^{13}CO_2$ gradient independently from the $^{12}CO_2$. Thus even though the total $CO_2$ concentration increased with soil depth, meaning an upward diffusion of $^{12}CO_2$, the $^{13}CO_2$ gradient was the opposite due to $^{13}C$-enriched leaf litter leading to a downward diffusion of $^{13}CO_2$. Consequently this could lead to a misinterpretation of the pathways of subsoil $^{13}CO_2$ in tracer experiments. Furthermore, this effect should also be taken into consideration when interpreting $^{14}CO_2$ soil profile measurements as an indicator of the age of the mineralised SOC, as in other field studies

(e.g., Davidson et al., 2006; Davidson and Trumbore, 1995; Fierer et al., 2005; Gaudinsik et al., 2000). Downward diffusion of $^{14}CO_2$ might be an important factor for explaining the observed $^{14}CO_2$ profiles. If this downward diffusion is the case, the $^{14}CO_2$ gradient should not have a continuous decrease with soil depth since the $^{14}CO_2$ gradient is the driving factor for diffusion according to Eq. (3). In fact, $^{14}CO_2$ concentration at 30 cm depth in subsoil OB1 was greater than at 50 cm depth (Fig. 10), which in turn led to a downward diffusion of $^{14}CO_2$ from a depth of 30 cm to 50 cm. This might lead to a rejuvenation of the

$^{14}CO_2$ soil profile and to an underestimation of the mineralisation of old SOC in subsoils.



## 5    Conclusions

The gradient method allowed total soil respiration to be partitioned vertically along a soil profile. Most of the $CO_2$ (90 %) was produced in the topsoil (< 30 cm). However, the subsoil (> 30 cm), which contained 47 % of SOC stocks, accounted for 10 % of total soil respiration. This can be explained by a larger amount of stable SOC in subsoils as compared to topsoils.

However, the modern radiocarbon signature of $CO_2$ throughout the soil profiles indicated that mainly young carbon sources were being respired, such as from roots and root exudates and autotrophic respiration. The contribution of old SOC to subsoil $CO_2$ production was too small to significantly alter the $^{14}C$ concentrations in the soil atmosphere used to identify $CO_2$ sources. Furthermore, this study showed that the mineralisation of fresh litter-derived C only contributed to a small part of total soil respiration, underlining the importance of roots and the rhizosphere for subsoil $CO_2$ production.

*Author contributions.*    All the authors contributed to the design of the field measurements and PWD carried out the field measurements. Preparation of 14CO2 samples was performed by PWD and AW. Data analysis and modelling were performed by PWD. KK took the root samples and analysed them and provided the data. PWD took the lead in writing the manuscript, with contributions from all the co-authors.

*Competing interests.*    The authors declare that they have no conflict of interest.

*Acknowledgements.*    This study was funded by the Deutsche Forschungsgemeinschaft (DFG) (HE 6877/1-1) within the framework of the
research unit SUBSOM (FOR1806) – "The Forgotten Part of Carbon Cycling: Organic Matter Storage and Turnover in Subsoils". We would like to thank Jens Dyckmanns and Reinhard Langel from the Centre for Stable Isotope Research and Analysis at the University of Göttingen for $^{13}13CO_2$ measurements. We also want to thank Frank Hegewald and Martin Volkmann for their support in the field, especially changing the 23 kg heavy batteries in the subsoil observatories every month. We would also like to thank Ullrich Dettmann for his support with R. Last but not least, many thanks to Marco Gronwald, Cora Vos and Viridiana Alcantara for fruitful discussions and recommendations.





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





**Figure 1.** Soil profile measurements of temperature (a), volumetric water content (b) and $CO_2$ concentration for the three observatories (OB). White bars represent periods without measurements.





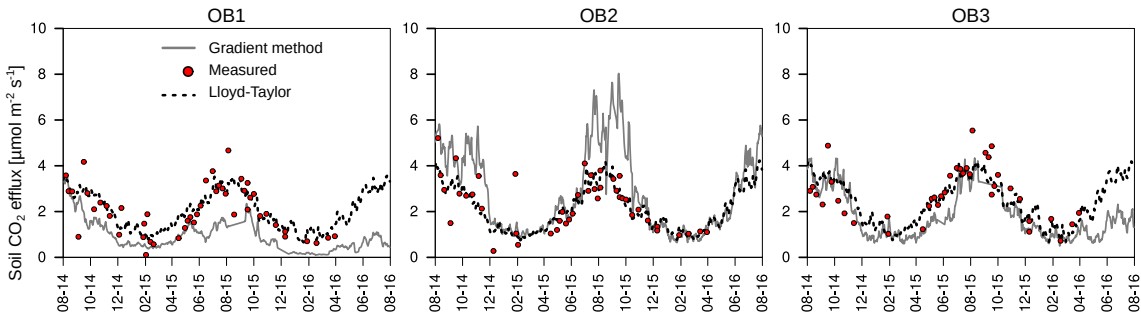

**Figure 2.** Mean daily soil respiration determined with the gradient method, measured with chambers and modelled with a Lloyd-Taylor function for the observatories (OB)





**Figure 3.** Daily mean $CO_2$ production in each soil layer (a)-(d). Arrows indicate disturbance due to bioturbation of voles in observatories (OB) 1 and 3, which created macropores and changed diffusivity.





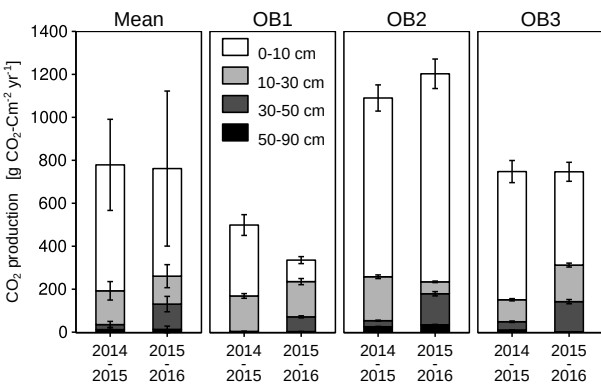

**Figure 4.** Cumulative $CO_2$ production for each soil layer, observatory (OB) and year of observation. Error bars represent standard deviation.

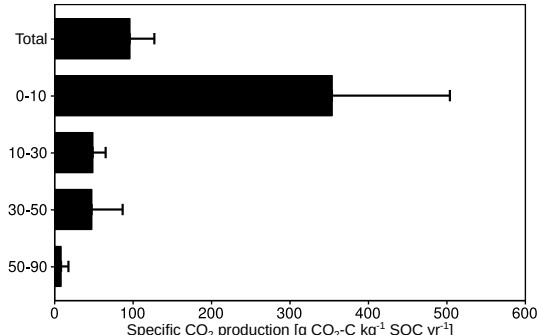

**Figure 5.** Mean annual specific $CO_2$ production for the total $CO_2$ efflux. Error bars represent standard deviation.



**Figure 6.** Isotopic signature of CO₂ at each depth and observatory (OB) before the addition of the labelled litter (a) and after labelled litter addition (b) with daily precipitation data (blue bars). The relative amount of litter-derived CO₂ on total CO2 in each depth and observatory (c). Please note the different y-axis ranges for (b) and (c).



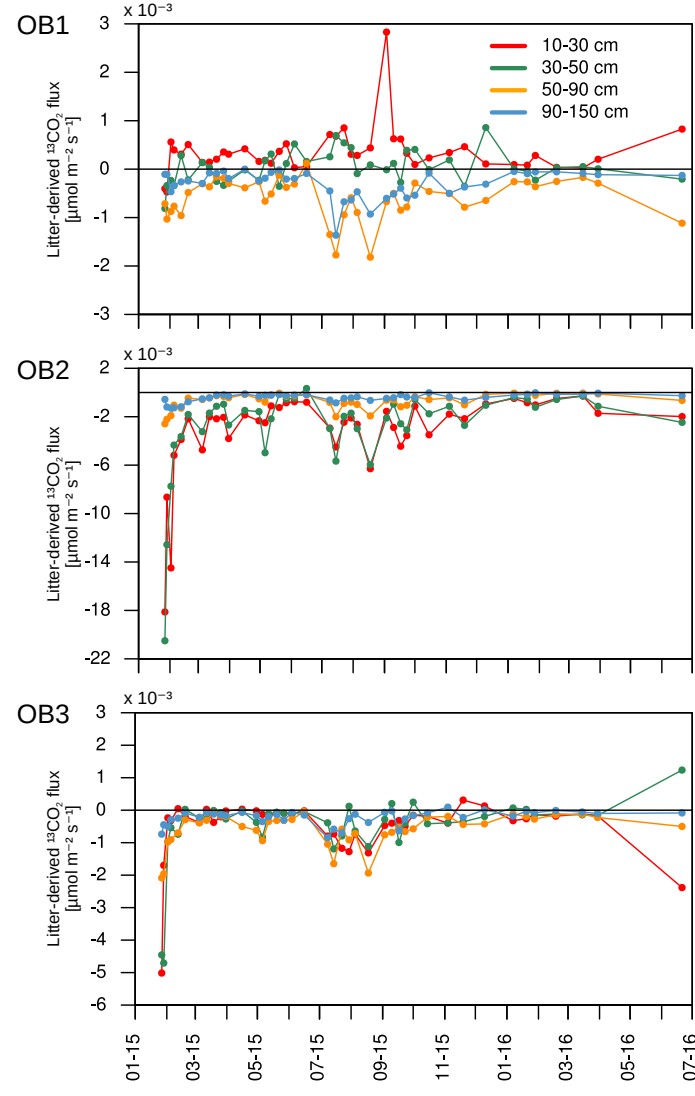

**Figure 7.** Litter-derived CO₂ fluxes for each observatory (OB). Positive fluxes represent mineralisation of litter-derived C. Negative fluxes represent diffusion from the layer above.



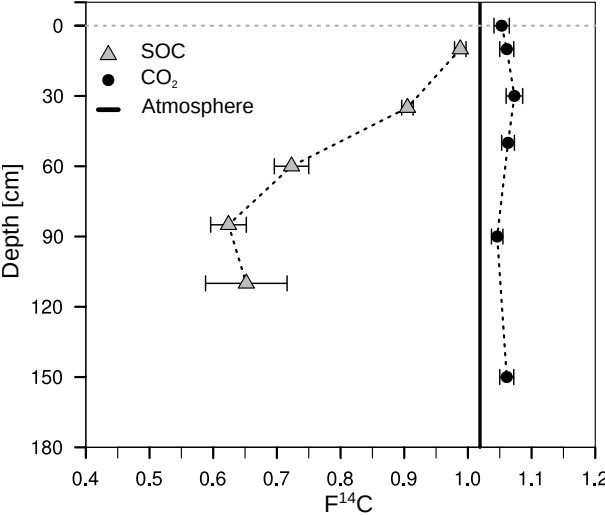

**Figure 8.** Mean $^{14}C$ concentration ($F^{14}C$) of bulk soil (grey triangles; data from Angst et al. (2016)) and $CO_2$ in the soil atmosphere (black dots). The solid black lines represents the annual average $F^{14}C$ value in the atmosphere from 2014 measured at the Jungfraujoch alpine research station, Switzerland (Levin and Hamer, pers. communication).

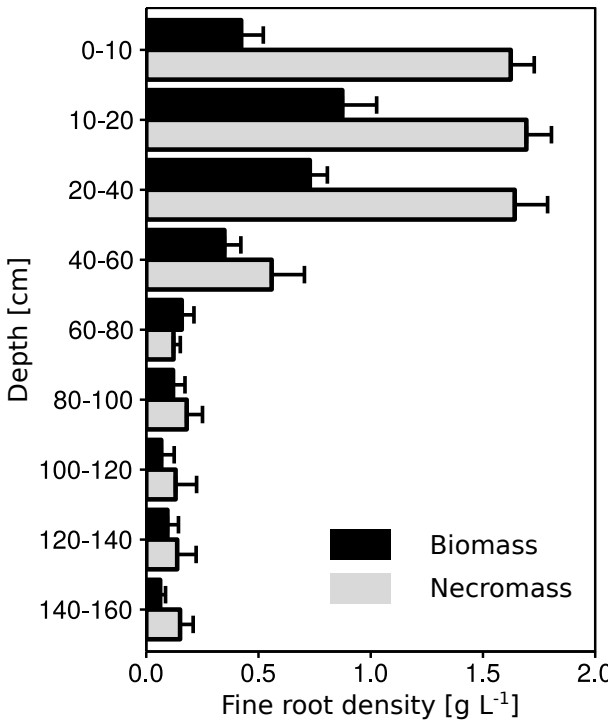

**Figure 9.** Mean fine root density for biomass and necromass of the subsoil observatories. Error bars represent standard error.





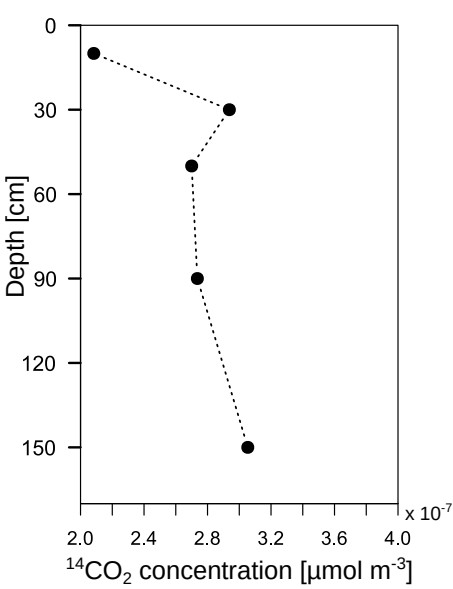

**Figure 10.** Soil air $^{14}CO_2$ concentration in observatory 1 from December 2014.





**Table 1.** Total soil respiration with and without the organic layer for the three observatories derived from soil surface measurements with linear interpolation (Chamber), modelled with a Lloyd-Taylor function and derived from the gradient method based on $CO_2$ measurements along the soil profile for one year. Means and standard deviations.

| Observatory | Soil respiration [g C m-2 yr-1] from August 2014 to August 2015 | | | | |
|---|---|---|---|---|---|
| | without organic layer | | | with organic layer | |
| | Chamber | Llyod-Taylor | Gradient method | Chamber | Llyod-Taylor |
| 1 | 699 (180) | 778 | 447 (54) | 923 (70) | 990 |
| 2 | 804 (211) | 780 | 1,080 (69) | 860 (273) | 816 |
| 3 | 824 (204) | 916 | 751 (56) | 1,120 (349) | 980 |
| Mean | 776 (193) | 825 (79) | 759 (317) | 967 (266) | 929 (98) |