# Peer review of "Vertical partitioning of CO2 production in a forest soil"

_Biogeosciences, 2019_

## Referee Comment (RC1) · Anonymous Referee #1 · 10 Jul 2019

This manuscript describes an investigation into sources of respiration of CO2 in soil profiles in a beech forest. I believe the measurements were generally done well and constitute a rare and valuable dataset, but I do have several questions and comments about the study setup and the data analysis that I would like to see addressed. I think if these concerns are addressed and data are reprocessed more appropriately (the production and isotope calculations in particular), this would be a unique and well-cited paper.

My understanding from reading the Methods several times is that Lysimeter tubes were temporarily installed in a beech tree forest soil, the soil inside was excavated away and used for sampling, and then the tubes were replaced with a large, solid polyethylene plug. The experimental treatments and measurements were then conducted in the area

immediately around the plugged hole. I may have this wrong, and I think a schematic figure showing the physical structure of the experimental setup would really help, or at a minimum clarifying the text. My initial impression at first reading was that the lysimeters remained and some experiments were done inside and others outside, which would have been very different.

The calculations of production rate and units were confusing (e.g. section 2.5.2, Fig's 3-5). I believe production needs to be expressed in conventional terms of unit volume, not area ($m^3$, not $m^2$). To calculate production with the gradient method, you need a difference between fluxes at two depths, and therefore must divide by the difference in depth, and end up with a unit volume in the denominator. You cannot use the gradient method to calculate production at a single depth because any horizontal plane with only two dimensions at some arbitrary soil depth only has one concentration gradient and one diffusivity, and so there is only one flux, and therefore zero production. When you want to sum the production at depth intervals to get the steady-state surface flux per unit area, you must multiply each each production value by the depth increment. If you apply your Eqn 8 to your modeled depths (without dividing) you would be comparing 10 cm to 40 cm depth intervals equally. Will you please clarify this?

Agreement between the profile method and the chamber measurements was off by quite a lot over large sections of time (Fig. 2), and could use more attention in the discussion. For OB1 and OB3 it looks like the modeled fluxes decrease relative to the surface fluxes over the course of the experiment. Could it possibly be that the flux gradient measurement area was impacted by the lysimeter installation (e.g. severed roots) in ways the surface fluxes were not? In OB2 the gradient method overestimated the flux during the growing season, possibly due to incorrect paramaterization of the model/diffusivity?

Why are there missing periods in the $CO_2$ profile data (Fig. 1c) but not in the flux gradient model results (Fig. 2)? Was there gap filling of some kind?

Decreasing concentrations with depth from 30-50 cm does not indicate that production was highest at these depths (section 4.1), but indicates non-steady state diffusion, possibly due to very wet soil and very low diffusivity (Fig. 1b). There would be a very slow net downward flux below these depths, but the source could still be the surface. Similarly, the "downward diffusion of 13CO2" after litter addition is a non-steady state phenomenon, and has nothing to do with the total CO2 gradient (Abstract, section 4.4.3).

For the isotope calculations, it appears you report the effect of label additions on delta-13C of CO2 at different depths. If I am mistaken about this I apologize and please clarify this in the text, but in Eqn 9, delta-13CM refers to a "gas sample", and Fig. 6c presents "litter-derived CO2". The isotope ratio of CO2 at a given depth does not tell you much of anything about production. It completely ignores the physics of diffusion.

Instead, the authors should calculate the isotope ratio of production at different depths (apply the gradient method to each isotopologue), or of the cumulative soil profile (Keeling method). For the gradient method, you would have to calculate fluxes and production of 12CO2 and 13CO2 separately throughout the profile, and then calculate the isotope ratio of production for each zone using the ratios of 13CO2 and 12CO2 produced per unit time: ((prod-13CO2/prod-12CO2)/R-VPDB)-1)*1000 per mil

Alternatively, you can use a Keeling plot approach for the whole profile), with a diffusion offset of 4.4 per mil on the offset to calculate the production signature of the entire profile (using all depths, so does not give information within the profile).

Then, after either of these, to know percent of label you would want to compare labelled and unlabelled plots over time to have the unlabelled endmember for a 2 source mixing model (use these values in equation 9 instead of the gas sample value). But, since there are no unlabelled plots, you will have to use the average or seasonal values from pre-treatment and state that you assume it would not have changed.

I believe the surface litter removal experiment would greatly underestimate the contri-
bution of litter to CO2 production. The insertion depth was 5 cm and the diameter of the chamber was 10.4 cm. The unsaturated layer of soil is at least two meters deep, and the CO mole fraction is tens of thousands of ppm at relatively shallow depths (Fig1c). Molecules of CO2 are moving in all directions under the soil and reflecting back off the lower boundary. Therefore, the volume of soil affecting the measurement made by the chamber is much larger than the volume of soil within the collar, and you would have to remove litter from a much larger area to see the effect in a surface flux measurement. For the same reason, it would be good to know the treatment area for the isotope-labelled litter addition. If the treatment area is small relative to the depth of the soil, the signal will disperse like a drop of ink into the ocean.

Lastly, I would consider changing the title to remove "in a dystric cambisol" and maybe instead using words that are more broadly relevant to raise the reach of the paper. If the soil type is important enough to put in the title, then I think there should be more text in the paper explaining the importance of the soil type for the contribution of this paper.

There is an impressive amount of CO2 profile and isotope profile data in this paper, and clearing up these analysis questions will make this a highly citable paper.

---

## Referee Comment (RC2) · Anonymous Referee #2 · 3 Aug 2019

The present study investigated the contribution of fresh litter-derived C to $CO_2$ production in the three soil profiles, the design and the methodology adopted was adequate, and the MS. is well written. However, the contribution of new C to $CO_2$ emissions can't be fully assessed by the 13C labelling experiment. And the conclusion of the importance of roots and the rhizosphere for $CO_2$ production, should be evidenced by input of labelled root or root exudate analog in additional treatments. This study is a two-year experiment. How to reduce the cross-feeding effect? Especially, the young beech litter can be assimilated into microbial biomass C. Did the formulas already take into account the cross-feeding effects between different C decomposition stages?

---

## Author Comment (AC1) · 7 Oct 2019

1. Referee comment:

My understanding from reading the Methods several times is that Lysimeter tubes were temporarily installed in a beech tree forest soil, the soil inside was excavated away and used for sampling, and then the tubes were replaced with a large, solid polyethylene plug. The experimental treatments and measurements were then conducted in the area immediately around the plugged hole. I may have this wrong, and I think a schematic figure showing the physical structure of the experimental setup would really help, or at a minimum clarifying the text. My initial impression at first reading was that the lysimeters remained and some experiments were done inside and others outside, which would

have been very different.

Authors response:

Yes the reviewer is right that a schematic figure will be helpful. We will add the following two figures to the manuscript (Fig 1 and Fig 2) which clarify the experimental set up. In addition we will add 2-3 sentences in the method section where we explain the origin of the gas samples and that all measurements were done outside the subsoil observatory. The subsoil observatories contained the data logger and the power supply for the sensors as well as the endings of the stainless steel tubes of the gas samplers.

2. Referee comment:

The calculations of production rate and units were confusing (e.g. section 2.5.2, Fig's 3-5). I believe production needs to be expressed in conventional terms of unit volume, not area (mËE̦3, not mËE̦2). To calculate production with the gradient method, you need a difference between fluxes at two depths, and therefore must divide by the difference in depth, and end up with a unit volume in the denominator. You cannot use the gradient method to calculate production at a single depth because any horizontal plane with only two dimensions at some arbitrary soil depth only has one concentration gradient and one diffusivity, and so there is only one flux, and therefore zero production. When you want to sum the production at depth intervals to get the steady-state surface flux per unit area, you must multiply each each production value by the depth increment. If you apply your Eqn 8 to your modeled depths (without dividing) you would be comparing 10 cm to 40 cm depth intervals equally. Will you please clarify this?

Authors response:

We thank the reviewer for the comment and we agree that the CO2 production expressed per area is a bit unusual. However, in the literature we found both expression per unit volume as well as area-based units see e.g. (Gaudinski et al. 2000; Hirano 2005; Fierer et al. 2005; Davidson et al. 2006; Hashimoto et al. 2007). Since SOC

stocks are also reported on an area basis, we decided to stick with the expression of unit per area for the $CO_2$ production, which might be easier to understand for a broader audience. We assumed that the $CO_2$ production in a certain soil layer can be described as the difference between the flux at the top of the soil layer and at the top of the soil layer below (e.g. Gaudinski et al. 2000). E.g. to calculate the $CO_2$ production in 10-30 cm depth we calculated the $CO_2$ flux between the sensor in 10 cm and 30 cm depth, this would represent the flux leaving the soil layer. For the $CO_2$ flux entering the soil layer between 10-30 cm we used the $CO_2$ gradient between 30 and 50 cm depth. We are not sure if we understood the point you are making about the comparison of the different depth intervals. We don't see a problem by comparing different depth intervals, since we always name the specific depth interval.

3. Referee comment:

Agreement between the profile method and the chamber measurements was off by quite a lot over large sections of time (Fig. 2), and could use more attention in the discussion. For OB1 and OB3 it looks like the modeled fluxes decrease relative to the surface fluxes over the course of the experiment. Could it possibly be that the flux gradient measurement area was impacted by the lysimeter installation (e.g. severed roots) in ways the surface fluxes were not?

Authors response:

The decrease in the surface fluxes derived from the gradient method of OB1 and OB3 can be explained by bioturbation (voles) in OB1 and OB3, which occurred in the second year, as tried to explain in the last sentence of section 4.2 (p.11 l30 – p.12 l1-3) and Fig 3a. In order to make things more clear, we will rephrase this paragraph and highlight more the problems of bioturbation which changed diffusivity in the first 10 cm of OB1 and OB3. The area around the $CO_2$ sensors where not affected by lysimeter installation.

4. Referee comment:

In OB2 the gradient method overestimated the flux during the growing season, possibly due to incorrect paramaterization of the model/diffusivity?

Authors response:

Yes the reviewer is right, the parametrization of the used diffusivity model in 10 cm depth at observatory 2 overestimated the fluxes. We reprocessed the data by using a fixed parametrization (without a distribution of the power fit function) of the diffusivity model for the specific depth and observatory. The total fluxes changed from 1080 g C m-2 yr-1 to 847 g C m-2 yr-1. We will change all figures and tables and the respective values in the text. Furthermore, in the final manuscript we will remove the distribution of the Ds model in the calculations for all observatories and depths and instead use the fixed parametrization set for each depth and observatory. This change will be made to be consistent with the data processing. The used parametrization values will be part of the supplement. However, there is still an overestimation of CO2 flux at OB2 during the growing season. This could possibly be explained by the lower measured soil moisture during the growing season at OB2 in 10 cm depth. In addition, OB2 had the highest total porosity of all three observatories (51 % vs 46 % and 49 %). In consequence the diffusivity at OB2 is higher during the growing season. As discussed in section 4.2 the difference between chamber measurements and the gradient method must be attributed to the spatial resolution of the measurement. At each observatory soil respiration was measured at 5 spatial replicates with the chamber method. Therefore, chamber measurements accounted for the spatial variability in water content and CO2 concentration below the chamber. However, there was no spatial replicate for the gradient method at the observatories.

5. Referee comment:

Why are there missing periods in the CO2 profile data (Fig. 1c) but not in the flux gradient model results (Fig. 2)? Was there gap filling of some kind?

Authors response:

Thank you for pointing that out. The missing periods are also in figure 2. However these period are difficult to see, because they appear as a straight line. This is just a plotting issue of R. However, the figure 2, especially the graph of the flux gradient method will be changed. In consequence, missing periods will be better visible (similar to Fig 3).

6. Referee comment:

For the isotope calculations, it appears you report the effect of label additions on delta-13C of CO2 at different depths. If I am mistaken about this I apologize and please clarify this in the text, but in Eqn 9, delta-13CM refers to a "gas sample", and Fig. 6c presents "litter-derived CO2". The isotope ratio of CO2 at a given depth does not tell you much of anything about production. It completely ignores the physics of diffusion. Instead, the authors should calculate the isotope ratio of production at different depths (apply the gradient method to each isotopologue), or of the cumulative soil profile (Keel-ing method). For the gradient method, you would have to calculate fluxes and production of 12CO2 and 13CO2 separately throughout the profile, and then calculate the isotope ratio of production for each zone using the ratios of 13CO2 and 12CO2 produced per unit time: ((prod-13CO2/prod-12CO2)/R-VPDB)-1)*1000 per mil Alternatively, you can use a Keeling plot approach for the whole profile), with a diffusion offset of 4.4 per mil on the offset to calculate the production signature of the entire profile (using all depths, so does not give information within the profile). Then, after either of these, to know percent of label you would want to compare labelled and unlabelled plots over time to have the unlabelled endmember for a 2 source mixing model (use these values in equation 9 instead of the gas sample value). But, since there are no unlabelled plots, you will have to use the average or seasonal values from pre-treatment and state that you assume it would not have changed.

Authors response:

We are happy for this comment, because it points out a mistake in our calculation of

litter-derived CO2 fluxes. As written in the manuscript we multiplied Eq. 9 with the absolute CO2 concentration to distinguish between 12CO2 and 13CO2 and afterwards we calculated litter-derived C fluxes. However, as the reviewer mentioned this was wrong. Furthermore, we must first calculate the CO2 fluxes / production in the respective layers for each sampling time. Then we must apply Eq. 9 on the CO2 production to the amount of litter mineralisation in the certain layer. As a reference value we use the average delta value for each depth and observatory before the labelling experiment started assuming that it would not have changed. We tried the suggested calculation from the reviewer for each isotopologue, but the derived delta values based on that calculation was on average -50 ‰ with a range of -400 ‰ to 40 ‰ which seems not realistic when compared to SOC delta values of ‑26.5Â㉠We think the Keeling plot approach for our soil profile is not suitable, since the diffusion offset of 4.4 ‰ is more theoretical and different from our field data. As shown Fig. 6a the delta values of CO2 in all depths and observatories showed almost similar values around 24 ‰ and we could not observe a change with depth. In consequence, we used the calculation as described below to estimate the litter-derived CO2 production. Changes in the manuscript: • Fig. 6c – Change title to Litter-derived C in CO2 • Fig. 7 – will be replaced by absolute 13CO2 fluxes • 2.5.3 Isotopic composition (p.7 l.25 ff.) "To determine the contribution of labelled litter-derived C to CO2 (L) in the soil atmosphere we used the isotopic mixing equation (Eq. 9): where $\delta$13CM is the isotopic signature of the soil atmosphere after the labelling, $\delta$13CL is the isotopic signature of the labelled leaf litter (1241 ‰ for OB1 and 1880 ‰ OB2 and OB3) and $\delta$13CB is the average isotopic signature of the soil atmosphere for each observatory and depth before the labelled leaf litter, which wouldn't change. Mineralisation of litter-derived C in each layer was calculated by multiplying the amount of litter-derived C with total CO2 production in the specific layer. Absolute 13CO2 concentration was calculated with isotopic signature of the soil atmosphere. Afterwards, 13CO2 fluxes and productions were calculated using Eq. (2)-(8). To account for different effective diffusivities of 12CO2 and 13CO2, the effective diffusivity Ds for 13CO2 was adjusted according to Cerling at al. (1991)"

7. Referee comment:

I believe the surface litter removal experiment would greatly underestimate the contribution of litter to CO2 production. The insertion depth was 5 cm and the diameter of the chamber was 10.4 cm. The unsaturated layer of soil is at least two meters deep, and the CO mole fraction is tens of thousands of ppm at relatively shallow depths (Fig1c). Molecules of CO2 are moving in all directions under the soil and reflecting back off the lower boundary. Therefore, the volume of soil affecting the measurement made by the chamber is much larger than the volume of soil within the collar, and you would have to remove litter from a much larger area to see the effect in a surface flux measurement.

Authors response:

The contribution of the organic layer to total soil respiration is in the range as found in other studies. Litter-derived CO2 accounts for 9.4 % to 37 % on total soil respiration as reported from litter manipulation experiments (Bowden et al. 1993; Nadelhoffer et al. 2004; Kim et al. 2005; Sulzman et al. 2005). However, we agree with the reviewer that the litter removal in the collar might underestimate the contribution of litter-derived CO2. We will add a paragraph in the discussion section explaining the problem with the litter removal as already pointed out by the reviewer. Nevertheless, since our data fit in the range as reported in the literature it is still reasonable to report them in the paper even if we may underestimate the litter-derived CO2.

8. Referee comment:

For the same reason, it would be good to know the treatment area for the isotope-labelled litter addition. If the treatment area is small relative to the depth of the soil, the signal will disperse like a drop of ink into the ocean.

Authors response:

The treatment area of the labelled litter was 6.6 m2. This information will be in the schematic figure and also written in the method section as written above.

9. Referee comment:

Lastly, I would consider changing the title to remove "in a dystric cambisol" and maybe instead using words that are more broadly relevant to raise the reach of the paper. If the soil type is important enough to put in the title, then I think there should be more text in the paper explaining the importance of the soil type for the contribution of this paper.

Authors response:

We agree with the reviewer to remove the soil type in the title. The title will be changed to "Vertical partitioning of $CO_2$ production in a forest soil".
* * *
[Figure]

**Fig. 1.** Schematic presentation of the subsoil observatories and the installed sensors and the labelling experiment (a) side view of the subsoil observatory and (b) topview of the labelled and control area.

[Figure]

**Fig. 2.** Photograph of the used lysimeter vessel to drill the whole for the subsoil observatory.

[Figure]

**Fig. 3.** Photograph of the used polyethylene shaft inserted thereafter.

[Figure]

**Fig. 4.** Absolute 13CO2 flux for each observatory and depth. Positive fluxes represents an upward diffusion of 13CO2 and negative fluxes represents downwards diffusion of 13CO2

[Figure]

---

## Author Comment (AC2) · 7 Oct 2019

1. Referee comment:

The present study investigated the contribution of fresh litter-derived C to CO2 production in the three soil profiles, the design and the methodology adopted was adequate, and the MS. is well written. However, the contribution of new C to CO2 emissions can't be fully assessed by the 13C labelling experiment. And the conclusion of the importance of roots and the rhizosphere for CO2 production, should be evidenced by input of labelled root or root exudate analog in additional treatments.

Authors response: We thank the reviewer for the interesting comment, unfortunately there is no analog experiment which could show the importance of roots and roots

exudates to CO2 production in the soil profile. Therefore, we can only rely on other studies which investigated the contribution of root respiration to total soil respiration such as Högberg et al. (2001). Still this is an interesting question and should be investigated in future studies.

We will add the following to discussion section (4.3, p.12 l12) "Even if we are unable with our study to distinguish between autotrophic and heterotrophic respiration, the importance of autotrophic respiration to total soil respiration was investigated in a large scale girdling experiment by Högberg et al. (2001). In their study they reported that autotrophic respiration accounted for up to 54 % of total soil respiration. In consequence root-derived respiration should be higher in the topsoil than in the subsoil, due to the decreasing root bio- and necromass with increasing soil depth.

2. Referee comment:

This study is a two-year experiment. How to reduce the cross-feeding effect? Especially, the young beech litter can be assimilated into microbial biomass C. Did the formulas already take into account the cross-feeding effects between different C decomposition stages?

Authors response:

We are not sure if we understand the comment correctly, but we didn't account for cross-feeding effects in the calculations, since we assume this was not the aim of the study.

---

## Author Comment (AC3) · 7 Oct 2019

References cited in the response

Bowden RD, Nadelhoffer KJ, Boone RD, et al (1993) Contributions of aboveground litter, belowground litter, and root respiration to total soil respiration in a temperate mixed hardwood forest. Can J For Res 23:1402–1407. doi: 10.1139/x93-177

Davidson EA, Savage KE, Trumbore SE, Borken W (2006) Vertical partitioning of CO2 production within a temperate forest soil. Glob Chang Biol 12:944–956. doi: 10.1111/j.1365-2486.2006.01142.x

Fierer N, Chadwick OA, Trumbore SE (2005) Production of CO2 in soil profiles of a California annual grassland. Ecosystems 8:412–429. doi: 10.1007/s10021-003-0151-

y

Gaudinski JB, Trumbore SE, Davidson EA, Zheng S (2000) Soil carbon cycling in a temperate forest: radiocarbon-based estimates of residence times, sequestration rates and partitioning of fluxes. Biogeochemistry 51:33–69. doi: doi.org/10.1023/A:1006301010014

Hashimoto S, Tanaka N, Kume T, et al (2007) Seasonality of vertically partitioned soil $CO_2$ production in temperate and tropical forest. J For Res 12:209–221. doi: 10.1007/s10310-007-0009-9

Hirano T (2005) Seasonal and diurnal variations in topsoil and subsoil respiration under snowpack in a temperate deciduous forest. Global Biogeochem Cycles 19:n/a-n/a. doi: 10.1029/2004GB002259

Högberg P, Nordgren A, Buchmann N, et al (2001) Large-scale forest girdling shows that current photosynthesis drives soil respiration. Nature 411:789–792. doi: 10.1038/35081058

Kim H, Hirano T, Koike T, Urano S (2005) Contribution of litter $CO_2$ production to total soil respiration in two different deciduous forests. Phyt - Ann Rei Bot 45:385–388

Nadelhoffer KJ, Boone RD, Bowden RD, et al (2004) The DIRT Experiment: Litter and Root Influences on Forest Soil Organic Matter Stocks and Function. In: FOSTER DR, Aber JD (eds) Forests in time: the environmental consequences of 1000 years of change in New England. Yale University Press, New Haven, Conneticut, pp 300–315

Sulzman EW, Brant JB, Bowden RD, Lajtha K (2005) Contribution of aboveground litter, belowground litter, and rhizosphere respiration to total soil $CO_2$ efflux in an old growth coniferous forest. Biogeochemistry 73:231–256. doi: 10.1007/s10533-004-7314-6

---

## Author Response (AR1)

**Referee 1**

| | |
|---|---|
| Comment 1 | My understanding from reading the Methods several times is that Lysimeter tubes were temporarily installed in a beech tree forest soil, the soil inside was excavated away and used for sampling, and then the tubes were replaced with a large, solid polyethylene plug. The experimental treatments and measurements were then conducted in the area immediately around the plugged hole. I may have this wrong, and I think a schematic figure showing the physical structure of the experimental setup would really help, or at a minimum clarifying the text. My initial impression at first reading was that the lysimeters remained and some experiments were done inside and others outside, which would have been very different. |
| Authors response | Yes the reviewer is right that a schematic figure will be helpful. We will add the following two figures to the manuscript (Fig 1 and Fig 2) which clarify the experimental set up. In addition we will add 2-3 sentences in the method section where we explain the origin of the gas samples and that all measurements were done outside the subsoil observatory. The subsoil observatories contained the data logger and the power supply for the sensors    as well as the endings of the stainless steel tubes of the gas samplers. |
| Changes | Two photographs were added (Fig. 1) showing the installation of the subsoil observatories. Further, we added a schematic figure (Fig. 2) showing the installations of the sensors in the soil and the labelling experiment. |
| | |
| Comment 2 | The calculations of production rate and units were confusing (e.g. section 2.5.2, Fig's 3-5). I believe production needs to be expressed in conventional terms of unit volume, not area (m^3, not m^2). To calculate production with the gradient method, you need a difference between fluxes at two depths, and therefore must divide by the difference in depth, and end up with a unit volume in the denominator. You cannot use the gradient method to calculate production at a single depth because any horizontal plane with only two dimensions at some arbitrary soil depth only has one concentration gradient and one diffusivity, and so there is only one flux, and therefore zero production. When you want to sum the production at depth intervals to get the steady-state surface flux per unit area, you must multiply each each production value by the depth increment. If you apply your Eqn 8 to your modeled depths (without dividing) you would be comparing 10 cm to 40 cm depth intervals equally. Will you please clarify this? |
| Authors response | We thank the reviewer for the comment and we agree that the $CO_2$ production expressed per area is a bit unusual. However, in the literature we found both expression per unit volume as well as area-based units see e.g. (Gaudinski et al. 2000; Hirano 2005; Fierer et al. 2005; Davidson et al. 2006; Hashimoto et al. 2007). Since SOC stocks are also reported on an area basis, we decided to stick with the expression of unit per area for the $CO_2$ production, which might be easier to understand for a broader audience.
 We assumed that the $CO_2$ production in a certain soil layer can be described as the difference between the flux at the top of the soil layer and at the top of the soil layer below (e.g. Gaudinski et al. 2000). E.g. to calculate the $CO_2$ production in 10-30 cm depth we calculated the $CO_2$ flux between the sensor in 10 cm and 30 cm depth, this would represent the flux leaving the soil layer. For the $CO_2$ flux entering the soil layer between 10-30 cm we used the CO2 gradient between 30 and 50 cm depth. We are not sure if we understood the point you are making about the comparison of the |

| | different depth intervals. We don't see a problem by comparing different depth intervals, since we always name the specific depth interval. |
|---|---|
| Changes | On p.7 l.16 we added literature references, which used the same calculation of $CO_2$ production |

| | |
|---|---|
| Comment 3 | Agreement between the profile method and the chamber measurements was off by quite a lot over large sections of time (Fig. 2), and could use more attention in the discussion. For OB1 and OB3 it looks like the modeled fluxes decrease relative to the surface fluxes over the course of the experiment. Could it possibly be that the flux gradient measurement area was impacted by the lysimeter installation (e.g. severed roots) in ways the surface fluxes were not? |
| Authors response | The decrease in the surface fluxes derived from the gradient method of OB1 and OB3 can be explained by bioturbation (voles) in OB1 and OB3, which occurred in the second year, as tried to explain in the last sentence of section 4.2 and Fig 3a. In order to make things more clear, we will rephrase this paragraph and highlight more the problems of bioturbation which changed diffusivity in the first 10 cm of OB1 and OB3. The area around the $CO_2$ sensors where not affected by lysimeter installation. |
| Changes | On p.12 l.13-l.24 we added
For example, the higher soil respiration determined with the gradient method at OB2 and OB3 in summer (Fig. 4) is linked to lower soil moisture measured in 10 cm depth (Fig. 3b) and to higher total soil porosity (51 % OB2, 49 % OB3 vs. 46 % OB1). In consequence, the effective diffusivity (Eq. 4) is higher, resulting in higher fluxes. Further, the lower soil respiration of OB1 and OB3 in the second year determined with the gradient method was related to bioturbation of voles, which increased the diffusivity around the $CO_2$ sensors and leading to a lower $CO_2$ concentration in 10 cm depth, which in turn led to an underestimation of total soil respiration (Fig. 4) by the gradient method. |

| | |
|---|---|
| Comment 4 | In OB2 the gradient method overestimated the flux during the growing season, possibly due to incorrect paramaterization of the model/diffusivity? |
| Authors response | Yes the reviewer is right, the parametrization of the used diffusivity model in 10 cm depth at observatory 2 overestimated the fluxes. We reprocessed the data by using a fixed parametrization (without a distribution of the power fit function) of the diffusivity model for the specific depth and observatory. The total fluxes changed from 1080 g C m$^{-2}$ yr$^{-1}$ to 847 g C m$^{-2}$ yr$^{-1}$. We will change all figures and tables and the respective values in the text. Furthermore, in the final manuscript we will remove the distribution of the $D_s$ model in the calculations for all observatories and depths and instead use the fixed parametrization set for each depth and observatory. This change will be made to be consistent with the data processing. The used parametrization values will be part of the supplement.
However, there is still an overestimation of $CO_2$ flux at OB2 during the growing season. This could possibly be explained by the lower measured soil moisture during the growing season at OB2 in 10 cm depth. In addition, OB2 had the highest total porosity of all three observatories (51 % vs 46 % and 49 %). In consequence the diffusivity at OB2 is higher during the growing season. As discussed in section 4.2 the difference between chamber measurements and the gradient method must be attributed to the spatial resolution of the measurement. At each observatory soil |

| | respiration was measured at 5 spatial replicates with the chamber method. Therefore, chamber measurements accounted for the spatial variability in water content and $CO_2$ concentration below the chamber. However, there was no spatial replicate for the gradient method at the observatories. |
|---|---|
| Changes | We recalculated $CO_2$ fluxes and production rate for all observatories and depths. Therefore fig 4 – fig 7 and table 1 was adjusted to the new values. Further, the results in the text (numbers) were adjusted. |

| | |
|---|---|
| Comment 5 | Why are there missing periods in the CO2 profile data (Fig. 1c) but not in the flux gradient model results (Fig. 2)? Was there gap filling of some kind? |
| Authors response | Thank you for pointing that out. The missing periods are also in figure 2. However these period are difficult to see, because they appear as a straight line. This is just a plotting issue of R. |
| Changes | Missing periods are now visible in figure 4. |

| | |
|---|---|
| Comment 6 | For the isotope calculations, it appears you report the effect of label additions on delta-13C of CO2 at different depths. If I am mistaken about this I apologize and please clarify this in the text, but in Eqn 9, delta-13CM refers to a "gas sample", and Fig. 6c presents "litter-derived CO2". The isotope ratio of CO2 at a given depth does not tell you much of anything about production. It completely ignores the physics of diffusion. Instead, the authors should calculate the isotope ratio of production at different depths (apply the gradient method to each isotopologue), or of the cumulative soil profile (Keel-ing method). For the gradient method, you would have to calculate fluxes and production of 12CO2 and 13CO2 separately throughout the profile, and then calculate the isotope ratio of production for each zone using the ratios of 13CO2 and 12CO2 produced per unit time: ((prod-13CO2/prod-12CO2)/R-VPDB)-1)*1000 per mil Alternatively, you can use a Keeling plot approach for the whole profile), with a diffusion offset of 4.4 per mil on the offset to calculate the production signature of the entire profile (using all depths, so does not give information within the profile). Then, after either of these, to know percent of label you would want to compare labelled and unlabelled plots over time to have the unlabelled endmember for a 2 source mixing model (use these values in equation 9 instead of the gas sample value). But, since there are no unlabelled plots, you will have to use the average or seasonal values from pre-treatment and state that you assume it would not have changed. |
| Authors response | We are happy for this comment, because it points out a mistake in our calculation of litter-derived $CO_2$ fluxes. As written in the manuscript we multiplied Eq. 9 with the absolute $CO_2$ concentration to distinguish between $^{12}CO_2$ and $^{13}CO_2$ and afterwards we calculated litter-derived C fluxes. However, as the reviewer mentioned this was wrong. Furthermore, we must first calculate the $CO_2$ fluxes / production in the respective layers for each sampling time. Then we must apply Eq. 9 on the $CO_2$ production to the amount of litter mineralisation in the certain layer. As a reference value we use the average delta value for each depth and observatory before the labelling experiment started assuming that it would not have changed.
We tried the suggested calculation from the reviewer for each isotopologue, but the derived delta values based on that calculation was on average -50 ‰ with a range of -400 ‰ to 40 ‰ which seems not realistic when compared to SOC delta values of -26.5 ‰. We think the Keeling plot approach for our soil profile is not suitable, since |

| | |
|---|---|
| | the diffusion offset of 4.4 ‰ is more theoretical and different from our field data. As shown Fig. 8a the delta values of $CO_2$ in all depths and observatories showed almost similar values around 24 ‰ and we could not observe a change with depth. In consequence, we used the calculation as described below to estimate the litter-derived $CO_2$ production. |
| Changes | • We rephrased the section 2.5.3 Isotopic composition of $CO_2$, accounting for the mistake in the calculation.
• We recalculated litter-derived C in $CO_2$ and added figure 9 and figure 10
• p.10 l.28 added "The total amount of labelled litter-derived C to the $CO_2$ production below 10 cm was 408 mg C m$^{-2}$ (± 329) (Fig. 9), which accounted for 0.18 % of total $CO_2$ production below 10 cm depth."
• p.10 l.30 – p.11 l.5 was rephrased, according to results from recalculation
• Fig. 8c changed title to "Litter-derived C in $CO_2$"
• Fig. 8c changed y axis label to "Amount of litter-derived $CO_2$ [%]"
• Replaced old figure 7 by figure 9 "Litter-derived $CO_2$ production"
• added figure 10 showing $^{13}CO_2$ fluxes |
| Comment 7 | I believe the surface litter removal experiment would greatly underestimate the contribution of litter to CO2 production. The insertion depth was 5 cm and the diameter of the chamber was 10.4 cm. The unsaturated layer of soil is at least two meters deep, and the CO mole fraction is tens of thousands of ppm at relatively shallow depths (Fig1c). Molecules of CO2 are moving in all directions under the soil and reflecting back off the lower boundary. Therefore, the volume of soil affecting the measurement made by the chamber is much larger than the volume of soil within the collar, and you would have to remove litter from a much larger area to see the effect in a surface flux measurement. |
| Authors response | The contribution of the organic layer to total soil respiration is in the range as found in other studies. Litter-derived $CO_2$ accounts for 9.4 % to 37 % on total soil respiration as reported from litter manipulation experiments (Bowden et al. 1993; Nadelhoffer et al. 2004; Kim et al. 2005; Sulzman et al. 2005). However, we agree with the reviewer that the litter removal in the collar might underestimate the contribution of litter-derived CO2. We will add a paragraph in the discussion section explaining the problem with the litter removal as already pointed out by the reviewer. Nevertheless, since our data fit in the range as reported in the literature it is still reasonable to report them in the paper even if we may underestimate the litter-derived $CO_2$. |
| Changes | added p.12 l.19-l.24
Removing the organic layer in the soil collars was supposed to determine the contribution of $CO_2$ production in the organic layer to total soil respiration. Since the organic layer was only removed in the soil collars and not around the soil collars, it must be noted that the contribution of the organic layer to total soil respiration might be underestimated with the used method. However, the results are in line with findings from litter manipulation experiments, which reported a contribution of 9 \% to 37 \% of the organic layer to total soil respiration (Nadelhoffer et al., 2004; Bowden et al., 1993; Kim et al., 2005; Sulzman et al., 2005). |
| Comment 8 | For the same reason, it would be good to know the treatment area for the isotope-labelled litter addition. If the treatment area is small relative to the depth of the soil, the signal will disperse like a drop of ink into the ocean. |

| | |
|---|---|
| Authors response | The treatment area of the labelled litter was 6.6 m$^2$. |
| Changes | We added figure 2 and added the information of the labelled area in the method section (p. 5 l. 22) |
| | |
| Comment 9 | Lastly, I would consider changing the title to remove "in a dystric cambisol" and maybe instead using words that are more broadly relevant to raise the reach of the paper. If the soil type is important enough to put in the title, then I think there should be more text in the paper explaining the importance of the soil type for the contribution of this paper. |
| Authors response | We agree with the reviewer to remove the soil type in the title. |
| Changes | The title was changed to "Vertical partitioning of $CO_2$ production in a forest soil" |

**Referee 2**

| | |
|---|---|
| Comment 1 | The present study investigated the contribution of fresh litter-derived C to CO2 production in the three soil profiles, the design and the methodology adopted was adequate, and the MS. is well written. However, the contribution of new C to CO2 emissions can't be fully assessed by the 13C labelling experiment. And the conclusion of the importance of roots and the rhizosphere for CO2 production, should be evidenced by input of labelled root or root exudate analog in additional treatments |
| Authors response | We thank the reviewer for the interesting comment, unfortunately there is no analog experiment which could show the importance of roots and roots exudates to $CO_2$ production in the soil profile. Therefore, we can only rely on other studies which investigated the contribution of root respiration to total soil respiration such as Högberg et al. (2001). Still this is an interesting question and should be investigated in future studies |
| Changes | We added on p.12 l.34 – p.13 l.4
Even if the current study is unable to distinguish between autotrophic and heterotrophic respiration, the importance of autotrophic respiration to total soil respiration was shown in a large scale girdling experiment by Högberg et al. (2001). They reported that autotrophic respiration accounted for up to 54 % on total soil respiration. In consequence, autotrophic respiration should be higher in the topsoil than in the subsoil, due to the decreasing root bio- and necromass with increasing soil depth (Fig. 12). |
| | |
| Comment 2 | This study is a two-year experiment. How to reduce the cross-feeding effect? Especially, the young beech litter can be assimilated into microbial biomass C. Did the formulas already take into account the cross-feeding effects between different C decomposition stages? |
| Authors response | We are not sure if we understand the comment correctly, but we didn't account for cross-feeding effects in the calculations, since this was not the aim of the study. |

Changes

[revised manuscript text omitted]

---

## Author Response (AR2)

Referee 1

| | |
|---|---|
| Comment 1 | I have revised MS " Vertical partitioning of CO2 production in a forest soil". The MS was much improved, compared to the previous version, however, I have a few additional comments for a final revision. This is a nice dataset and study.
In Fig.8b, please break the X-axis, so that it's better to compare the 13C abundances between before labeling and after labeling. |
| Authors response | Figure 8b was changed. A break at -20 ‰ was added. We assume the referee means the y and not the x-axis. |
| Changes | A break of the Y-axis was added in Figure 8b. |
| Comment 2 | The labeling experiment was performed in January 2015, and the labeled beech litter was used at the beginning of this experiment, therefore, the $^{13}C-CO_2$ might not directly derive from the litter but from microbial biomass turnover after one or two years later, which overestimated the fresh litter-derived C contributed to $CO_2$ production. |
| Authors response | We thank the referee for this comment. We can see the point that turnover of microbial biomass may interfere with the isotopic signature of the soil atmosphere. However, we think that the error introduced by neglecting microbial biomass turnover on the isotopic signature of $CO_2$ is small. Microbial biomass makes up only a small part of total SOC. In addition, not all microbial biomass is mineralised. Further, the isotopic signature of microbial biomass ranges from is -27 ‰ in 10 cm -25.8 ‰ in 90 cm at the study site (pers. communication S. Preußer), which is even lower than the isotopic signature of soil $CO_2$ before the application of the labelled leaf litter.
Therefore, the isotopic values of the microbial biomass and the small contribution of microbial biomass turnover to $CO_2$ production, indicates that the turnover of old biomass has no measurable effect on the isotopic ratio of the $CO_2$ in the soil atmosphere. We have added this point to the discussion section. |
| Changes | • p.13 l.26 l.30 added:
    ○ It should be considered that part of the measured $^{13}CO_2$ may derives from the turnover of the microbial necromass, which could led to an overestimation of the litter-derived $CO_2$. However, the isotopic signature of the biomass at the study site ranges from -27 ‰ (10 cm) to -25.8 ‰ (90 cm) (Preußer and Kandeler pers. communication) which is lower than the isotopic signature of the soil atmosphere before the application of the labelled leaf litter. This indicates that the turnover of microbial necromass had no measurable effect on the isotopic signature of the soil atmosphere. |

Referee 2

| | |
|---|---|
| Comment 1 | I understand the challenges of determining isotope ratios of production with these data, and appreciate the efforts of the authors after my initial comment. Even for the unlabeled background case, the Keeling plot method is difficult if you do not have measurements very close to the surface where most of the isotope variation occurs. If all of your depths are around -24 to -25 per mil, then your overall source for the profile is probably very close to -29. The theoretical diffusive enrichment of 4.4 has been confirmed in many field studies. Given the large difference of isotope ratio of the litter (1241 and 1880 per mil), -29 per mil is probably a close enough approximation for your background endmember.

 For determining amount of label in production, I still do not believe the paper gets it right, and I think the underlying problem is in trying to apply steady-state calculations to the non-steady state conditions induced by the label addition. I think the best way to interpret these isotopic data would be a non-steady-state model, which would be capable of accounting for the slow equilibration apparent in Fig 8b. After the litter addition, the label-derived CO2 slowly makes its way down the profile, while simultaneously diffusing out the top. Perhaps this effort could warrant a separate paper. |
| Authors response | We thank the reviewer for this critical comment. We see the point that a non-steady-state model would be better for the interpretation of the observed isotopic data. However, we are not sure if a non-steady-state model will work with the current methods, especially the low temporal resolution of the isotopic data. In order to take into account the diffusion from the interface organic layer to mineral soil, a higher spatial resolution of measurement points for $CO_2$, $^{13}CO_2$ and water content between 0 and 10 cm depth would be necessary. In addition, a diffusion model is also required for the organic layer and the transition from the organic layer to the mineral soil. We agree with the reviewer that another paper should address this interesting issue. For now, we sticked with the steady-state model, but addressed the critical points in our discussion (please see also our next comment).

 The positive $^{13}CO_2$ fluxes between 10 and 30 cm suggest (Fig 10) that litter-derived C was transported as DOM down the soil profile and mineralised in deeper horizons. From the used methods and measured data, we believe the presented analysis gave us a "good" approximation on the contribution of litter-derived C to $CO_2$ production with in deeper soil horizons, even with the uncertainties introduced by the used steady-state model. |
| Comment 2 | I agree with the authors' response where they said, "Furthermore, we must first calculate the $CO_2$ fluxes / production in the respective layers for each sampling time. Then we must apply Eq. 9 on the $CO_2$ production to [obtain] the amount of litter mineralisation in the certain layer." However, it appears in section 2.5.3 that the authors are still applying equation 9 to soil $CO_2$, which is not valid, and then combining this "scaling factor" with total $CO_2$ production calculated for each layer. Amount of label C in soil $CO_2$, does not really tell you much about production by itself. If production rate or diffusivity have changed, the comparison goes out the window. If the authors were able to calculate $^{13}CO_2$ fluxes by layer, then they can do it for $^{12}CO_2$ as well, and then can apply equation 9 correctly to the $CO_2$ produced within each layer (as stated in the reply, but not changed in the manuscript). If this does not produce reasonable results, it is probably due to a violation of steady state assumptions, requiring a non-steady state model. |
| Authors response | We see the critical point with the non-steady-state conditions and the derived isotopic values of the $^{13}CO_2$ production as mentioned in the first response letter. |

In the first response we had decided against the proposed calculation because of the sometimes unrealistic isotope values of the $CO_2$-production (e.g. -400 ‰). However, most of these unrealistic values occurred at very low CO2 production rates < 0.01 µmol m$^{-2}$ s$^{-1}$. We changed the calculation as suggested by the reviewer, but the change in the amount and the contribution of litter-derived CO2 production to total CO2 production was minor. The average amount of litter-derived CO2 below 10 cm changed from 408 mg C m-2 to 291 mg C m-2 (January 2015 to June 2016).

| | |
|---|---|
| Changes | In section 2.5.3 we added the necessary information for the calculation of litter-derived $CO_2$ production. |

- p.8 l.8 added:
  - The $CO_2$ fluxes and productions for each layer and isotopologue of $CO_2$ (12CO2 and 13CO2) were calculated using the isotopic signature of the soil atmosphere and Eq. (2-7).
- p.8 l.13 - l.20 added:
  - To determine the contribution of the labelled leaf litter to the $CO_2$ production in a soil layer and accounting for diffusion effects, the isotopic signature of $CO_2$ production ($\delta^{13}P$-$CO_2$) in each soil layer was calculated with Eq. 11.
  - Eq. 11 was added
  - where $R_{st}$ is the isotopic ratio of the Vienna-PDB reference standard, while $^{13}P$-$CO_2$ and $^{12}P$-$CO_2$ are the $CO_2$ production for each isotopologue of the respective soil layer. Afterwards, Eq. 9 was used to calculate the amount of labelled leaf litter to total $CO_2$ production, where $\delta^{13}C_B$ was substituted with the average isotopic signature of $CO_2$ production (Eq. 11) before the labelling and $\delta^{13}C_M$ was substituted with the isotopic signature of $CO_2$ production. The litter-derived $CO_2$ production was calculated by multiplying the amount of labelled leaf litter (L) with the total $CO_2$ production of the respective soil layer.
- p.11 l2 values for litter-derived CO2 were changed
  - from 408 mg C m$^{-2}$ (± 329) to from 291 mg C m$^{-2}$ (± 127)
  - from 0.18 % to 0.12 %
- p. 13 l. 25 values for annual litter-derived $CO_2$ were changed
  - from 0.12 to 0.13 %
- p.13 l. 30 – p.14 l.5 were added

[revised manuscript text omitted]

---

## Author Response (AR3)

Authors response:

Minor changes were made in the acknowledgments and the bibliography was cleaned up
- l. 445 – l. 446 added
  We acknowledge support by the Open Access Publication Funds of the SLUB/TU Dresden for financing the open access publication.
- l.450 deleted "comments"
- l. 452 – l. 574 the url was removed from the references, because the doi already contains the link to the reference.
- l. 59 and l. 508-510 reference was added

[revised manuscript text omitted]